# PrivacyGLUE: A Benchmark Dataset for General Language Understanding in Privacy Policies

Atreya Shankar *,† , Andreas Waldis † , Christof Bless † , Maria Andueza Rodriguez † and Luca Mazzola *,†

Information Systems Research Lab, HSLU—Lucerne University of Applied Sciences and Arts, Suurstoffi 1,
CH-6343 Rotkreuz, Switzerland
* Correspondence: atreya.shankar@hslu.ch (A.S.); luca.mazzola@hslu.ch (L.M.);
  Tel.: +41-41-757-30-97 (A.S.); +41-41-757-68-90 (L.M.)
† These authors contributed equally to this work.

**Featured Application: We propose the PrivacyGLUE benchmark to compare and contrast NLP models' general language understanding in the privacy language domain. This will help practitioners in selecting understanding models for applications within the privacy language domain.**

**Abstract:** Benchmarks for general language understanding have been rapidly developing in recent years of NLP research, particularly because of their utility in choosing strong-performing models for practical downstream applications. While benchmarks have been proposed in the legal language domain, virtually no such benchmarks exist for privacy policies despite their increasing importance in modern digital life. This could be explained by privacy policies falling under the legal language domain, but we find evidence to the contrary that motivates a separate benchmark for privacy policies. Consequently, we propose PrivacyGLUE as the first comprehensive benchmark of relevant and high-quality privacy tasks for measuring general language understanding in the privacy language domain. Furthermore, we release performances from multiple transformer language models and perform model–pair agreement analysis to detect tasks where models benefited from domain specialization. Our findings show the importance of in-domain pretraining for privacy policies. We believe PrivacyGLUE can accelerate NLP research and improve general language understanding for humans and AI algorithms in the privacy language domain, thus supporting the adoption and acceptance rates of solutions based on it.

**Keywords:** privacy policies; NLP; benchmark; general language understanding; domain specialization and generalization



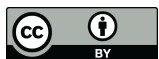

## 1. Introduction

Data privacy is evolving into a critical aspect of modern life, with the United Nations (UN) describing it as a *human right in the digital age* [1]. Data privacy practices are often disclosed in complex legal documents known as privacy policies, and are commonly encountered in daily digital life when visiting websites or utilizing online services. Therefore, the comprehension of privacy policies is important as it strongly correlates with the comprehension of one's data privacy. Despite this importance, several studies have demonstrated high barriers to the understanding of privacy policies due to their length and legal jargon [2]. McDonald and Cranor [3] estimate that an average person requires ∼200 h annually to read all privacy policies encountered in their daily life. The negative consequences of accepting privacy policies without comprehension could be significant. Obar and Oeldorf-Hirsch [2] demonstrated that most users in a survey mistakenly consented to *gotcha* clauses which enabled the sharing of their private data with intelligence authorities and employers. Solutions to the problem of privacy policy comprehension are in active discussion, with studies such as Wilson et al. [4] advocating for the training of Artificial Intelligence (AI) algorithms on appropriate benchmark datasets to assist humans in understanding privacy policies.

In recent years, benchmarks have been gaining popularity in Machine Learning and Natural Language Processing (NLP) communities because of their ability to holistically evaluate model performance over a variety of representative tasks, thus allowing practitioners to compare and contrast different models on multiple tasks relevant for the specific application domain. General Language Understanding Evaluation (GLUE) [5] and Super-GLUE [6] are examples of popular NLP benchmarks which measure the natural language understanding capabilities of state-of-the-art (SOTA) models. NLP benchmarks are also developing rapidly in language domains, with LexGLUE [7] being an example of a recent benchmark hosting several difficult tasks in the legal language domain. Interestingly, we do not find similar NLP benchmarks in the privacy language domain for privacy policies. While this could be explained by privacy policies falling under the legal language domain due to their formal and jargon-heavy nature, we claim that privacy policies fall under a distinct language domain and cannot be subsumed under any other specialized NLP benchmark such as LexGLUE.

To investigate this claim, we gather documents from Wikipedia [8], European Legislation (EURLEX) [9] and company privacy policies [10], with each corpus truncated to 2.5 M tokens. Next, we feed these documents into BERT and gather contextualized embeddings, which are then projected to a two-dimensional space using Uniform Manifold Approximation and Projection (UMAP) [11]. In Figure 1, we observe that the three domain corpora cluster independently, providing evidence that privacy policies lie in a distinct language domain from both legal and Wikipedia documents, and therefore require an independent NLP benchmark. With this motivation, we propose PrivacyGLUE as the first comprehensive benchmark for measuring general language understanding in the privacy language domain. Our main contributions are threefold:

1. Composition of seven high-quality and relevant PrivacyGLUE tasks, specifically OPP-115, PI-Extract, Policy-Detection, PolicyIE-A, PolicyIE-B, PolicyQA and PrivacyQA.
2. Benchmark performances of five transformer language models on all aforementioned tasks, specifically BERT, RoBERTa, Legal-BERT, Legal-RoBERTa and PrivBERT.
3. Model agreement analysis to detect PrivacyGLUE task examples where models benefited from domain specialization.

We release PrivacyGLUE as a fully configurable benchmark suite for straight-forward reproducibility and production of new results in our public GitHub repository: https://github.com/infsys-lab/privacy-glue (accessed on 22 January 2023).

We illustrate our methodologies in the form of a flowchart in Figure 2. Our findings show that PrivBERT, the only model pretrained on privacy policies, outperforms other models by an average of 2–3% over all PrivacyGLUE tasks, shedding light on the importance of in-domain pretraining for privacy policies. Our model–pair agreement analysis explores specific examples where PrivBERT's privacy-domain pretraining provided both a competitive advantage and disadvantage. By benchmarking holistic model performances, we believe PrivacyGLUE can accelerate NLP research into the privacy language domain and ultimately improve general language understanding of privacy policies for both humans and AI algorithms.

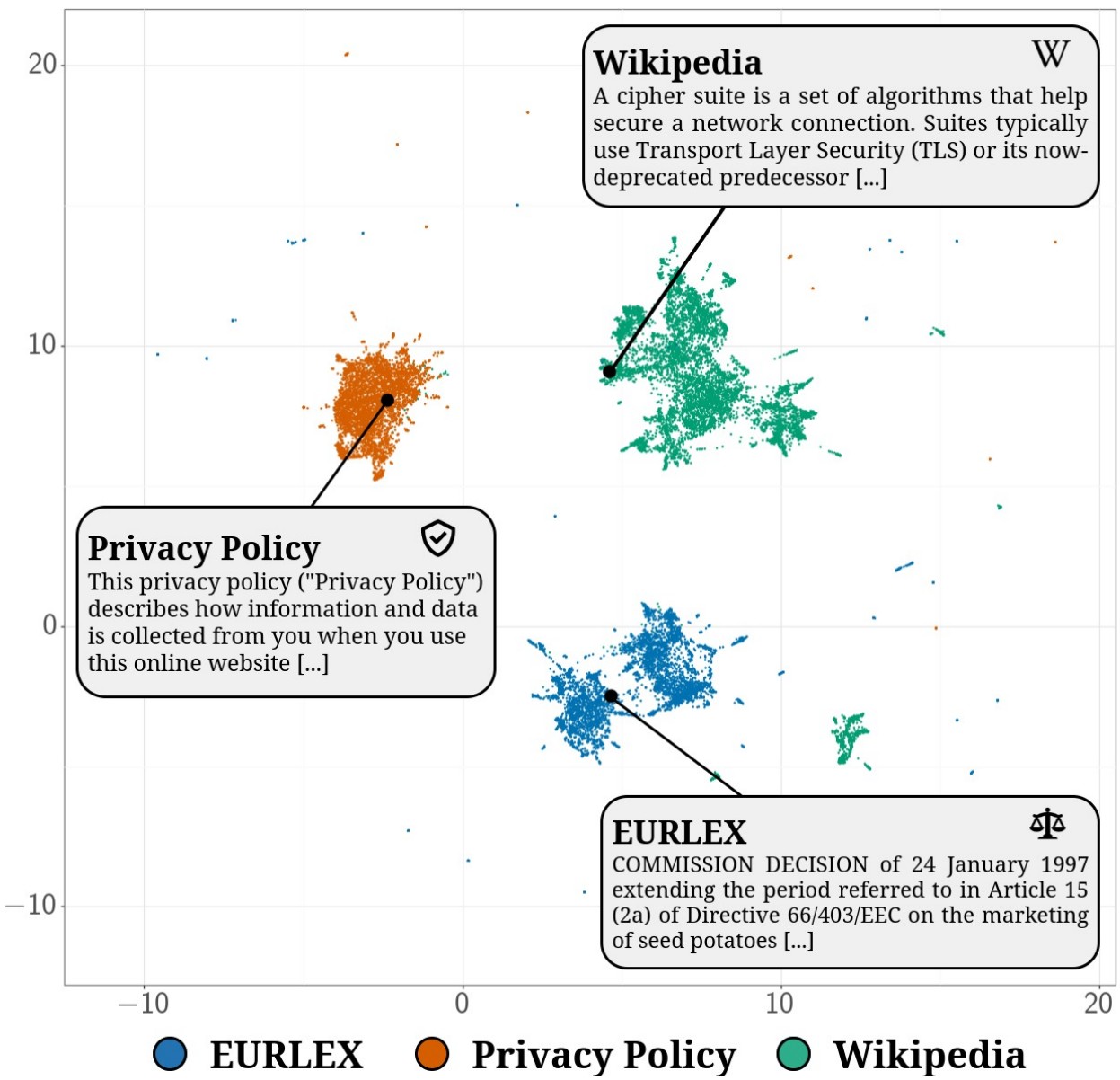

**Figure 1.** UMAP visualization of BERT embeddings from Wikipedia, European Legislation (EURLEX) and company privacy policy documents with a total of 2.5 M tokens per corpus.

Innovative aspects of our study include consulting European and American privacy experts to determine challenging task types where AI algorithms could assist humans. Based on this analysis, we carefully select high-quality and publicly available datasets originating from peer-reviewed studies. We refine these datasets by applying transformations to shape them into tasks that would be useful for improving comprehension of privacy policies to average users. Next, we selected pretrained models from various language domains and fine-tuned them on PrivacyGLUE tasks. Finally, we provide a model disagreement analysis to investigate samples where in-domain pretraining led to specialization and the lack thereof.

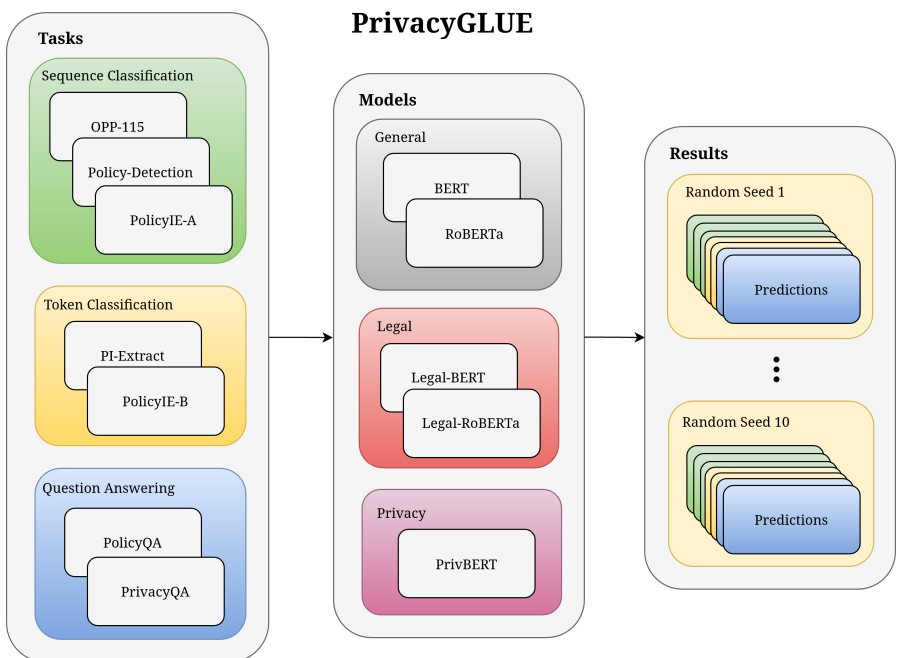

**Model Disagreement Analysis**

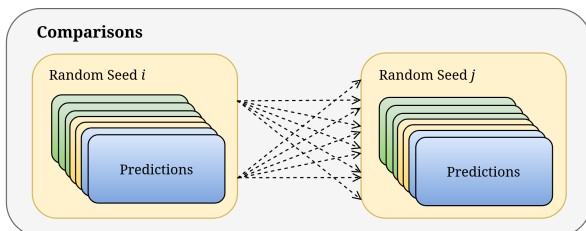

**Figure 2.** Flowchart depicting our main contributions, that is, the PrivacyGLUE benchmark with its tasks and models, along with the model disagreement analysis proposed in our study.

## 2. Related Work

NLP benchmarks have been gaining popularity in recent years because of their ability to holistically evaluate model performance over a variety of representative tasks. GLUE [5] and SuperGLUE [6] are examples of benchmarks that evaluate SOTA models on a range of natural language understanding tasks. The Generation, Evaluation and Metrics (GEM) benchmark [12] looks beyond text classification and measures performance in Natural Language Generation tasks, such as summarization and data-to-text conversion. The Cross-Lingual Transfer Evaluation of Multilingual Encoders (XTREME) [13] and Cross-Lingual Transfer Evaluation of Multilingual Encoders Revisited (XTREME-R) [14] benchmarks specialize in measuring cross-lingual transfer learning on 40–50 typologically diverse languages and corresponding tasks. Popular NLP benchmarks often host public leaderboards with SOTA scores on supported tasks, thereby encouraging the community to apply new approaches for surpassing top scores.

While the aforementioned benchmarks focus on problem types such as natural language understanding and generation, other benchmarks focus on language domains. The LexGLUE benchmark [7] is an example of a benchmark that evaluates models on tasks from the legal language domain. LexGLUE consists of seven English language tasks that are representative of the legal language domain and chosen based on size and legal specialization. Chalkidis et al. [7] benchmarked several models, such as Bidirectional Encoder Representations from Transformers (BERT) [15] and Legal-BERT [16], where Legal-BERT has a similar architecture to BERT but was pretrained on diverse legal corpora. A key finding of LexGLUE was that Legal-BERT outperformed other models which were not

pretrained on legal corpora. In other words, they found that an in-domain pretrained model outperformed models that were pretrained out-of-domain.

In the privacy language domain, we tend to find isolated datasets from specialized studies. The Refs. [4,17–19] are examples of studies that introduce annotated corpora for a privacy-practice sequence and token classification tasks, while the Refs. [20,21] release annotated corpora for privacy-practice question answering. Amos et al. [22] is another recent study that released an annotated corpus of privacy policies. As of writing, no comprehensive NLP benchmark exists for general language understanding in privacy policies, making PrivacyGLUE the first consolidated NLP benchmark in the privacy language domain.

## 3. Datasets and Tasks

During the composition phase of the PrivacyGLUE benchmark, we consulted European and American privacy experts on aspects of privacy policies that are particularly challenging for non-expert users to comprehend. We found these challenging aspects to be well-addressed by NLP models trained on the sequence classification, token classification and question-answering task types. We then searched for publicly available datasets in the privacy domain that fit our task type requirements. We refined our selection by keeping datasets that had at least ∼1K total samples of sufficient quality, prioritizing datasets that were accompanied by a peer-reviewed scientific study. With this, we composed the PrivacyGLUE benchmark using seven natural language understanding tasks originating from six datasets in the privacy language domain, which we describe in subsequent sections. Summary statistics, detailed label information and representative examples are shown in Table 1, Table A1 (Appendix A) and Table A2 (Appendix B), respectively.

**Table 1.** Summary statistics of PrivacyGLUE benchmark tasks; ‡ PI-Extract and PolicyIE-B consist of four and two subtasks, respectively, and the number of BIO token classes per subtask are separated by a forward-slash character.

| Task | Source | Task Type | Train/Dev/Test Instances | # Classes |
|------|--------|-----------|--------------------------|-----------|
| OPP-115 | Wilson et al. [4] | Multi-label sequence classification | 2185/550/697 | 12 |
| PI-Extract | Bui et al. [18] | Multi-task token classification | 2579/456/1029 | 3/3/3/3 ‡ |
| Policy-Detection | Amos et al. [22] | Binary sequence classification | 773/137/391 | 2 |
| PolicyIE-A | Ahmad et al. [19] | Multi-class sequence classification | 4109/100/1041 | 5 |
| PolicyIE-B | Ahmad et al. [19] | Multi-task token classification | 4109/100/1041 | 29/9 ‡ |
| PolicyQA | Ahmad et al. [21] | Reading comprehension | 17,056/3809/4152 | – |
| PrivacyQA | Ravichander et al. [20] | Binary sequence classification | 157,420/27,780/62,150 | 2 |

### 3.1. OPP-115

Wilson et al. [4] was the first study to release a large annotated corpus of privacy policies. A total of 115 privacy policies were selected based on their corresponding company's popularity on Google Trends. The selected privacy policies were annotated with 12 data privacy practices on a paragraph-segment level by experts in the privacy domain. As noted by Mousavi Nejad et al. [23], one limitation of Wilson et al. [4] was the lack of publicly released training and test data splits which are essential for machine learning and benchmarking. To address this, Mousavi Nejad et al. [23] released their own training, validation and test data splits for researchers to easily reproduce OPP-115 results. PrivacyGLUE utilizes the "Majority" variant of data splits released by Mousavi Nejad et al. [23] to compose the OPP-115 task. Given an input paragraph segment of a privacy policy, the goal of OPP-115 is to predict one or more data practice categories.

### 3.2. PI-Extract

Bui et al. [18] focus on enhanced data practice extraction and presentation to help users better understand privacy policies. As part of their study, they released the PI-Extract dataset consisting of 4.1 K sentences (97 K tokens) and 2.6 K expert-annotated data practices from 30 privacy policies in the OPP-115 dataset. Expert annotations were

performed on a token level for all sentences of selected privacy policies. The PI Extract was broken down into four subtasks, where spans of tokens were independently tagged using the Beginning, Inside and Outside (BIO) scheme commonly used in Named Entity Recognition (NER). Subtasks I, II, III and IV require the classification of token spans for data-related entities that are collected, not collected, not shared, and shared, respectively. In the interest of diversifying tasks in PrivacyGLUE, we composed PI-Extract as a multi-task token classification problem where all four PI-Extract subtasks are to be jointly learned.

### 3.3. Policy Detection

Amos et al. [22] developed a crawler for automated collection and curation of privacy policies. An important aspect of their system is the automated classification of documents into privacy policies and non-privacy-policy documents encountered during web-crawling. To train such a privacy policy classifier, Amos et al. [22] performed expert annotations of commonly encountered documents during web crawls and classified them into the aforementioned categories. The Policy Detection dataset was released with a total of 1.3 K annotated documents and is utilized in PrivacyGLUE as a binary sequence classification task.

### 3.4. PolicyIE

Inspired by the Refs. [4,18,19], PolicyIE was created, an English corpus composed of 5.3 K sentence-level and 11.8 K token-level data practice annotations over 31 privacy policies from websites and mobile applications. PolicyIE was designed to be used for machine learning in NLP, to ultimately make data privacy concepts easier for users to understand. We split the PolicyIE corpus into two tasks, namely PolicyIE-A and PolicyIE-B. Given an input sentence, PolicyIE-A entails multi-class data practice classification while PolicyIE-B entails multi-task token classification over distinct subtasks I and II, which require the classification of token spans for entities that participate in privacy practices and their conditions/purposes, respectively. The motivation for composing PolicyIE-B as a multi-task problem is similar to that of the PI Extract.

### 3.5. PolicyQA

Ahmad et al. [21] argue in favour of short-span answers to user questions for long privacy policies. They release PolicyQA, a dataset of 25 K reading comprehension examples curated from the OPP-115 corpus from Wilson et al. [4]. Furthermore, they provide 714 human-written questions optimized for a wide range of privacy policies. The final question–answer annotations follow the Stanford Question Answering Dataset (SQuAD) 1.0 format [24], which improves the ease of adaptation into NLP pipelines. We utilize PolicyQA as PrivacyGLUE's reading comprehension task.

### 3.6. PrivacyQA

Similar to [20,21] who argued in favour of annotated question-answering data for training NLP models to answer user questions about privacy policies, they correspondingly released PrivacyQA, a corpus composed by 1.75 K questions and more than 3.5 K expert-annotated answers from 35 privacy policies. Unlike PolicyQA, PrivacyQA proposes a binary sequence classification task where a question–answer pair is classified as either relevant or irrelevant. Correspondingly, we treat PrivacyQA as a binary sequence classification task in PrivacyGLUE.

## 4. Experimental Setup

The PrivacyGLUE benchmark was tested using the BERT, RoBERTa, Legal-BERT, Legal-RoBERTa and PrivBERT models, which are summarized in Table 2. We prioritized these models since they are of similar size but differ in terms of their pretraining corpora, which are the general, legal and privacy language domains, respectively. As a result, this selection would provide us with insights about the influence of pretraining on downstream

performance. In this section, we describe the models used and task-specific approaches, and provide details on our benchmark configuration.

### 4.1. Models

This section introduces the models which are currently available in our PrivacyGLUE benchmark release. Table 2 provides a synoptic, comparative view on the important properties of the models, while the next paragraphs introduce their origin, scope, and relevant literature pointers.

**Table 2.** Summary of models used in the PrivacyGLUE benchmark; all models used are base-sized variants of BERT/RoBERTa architectures; § BC = BookCorpus, CC-News = CommonCrawl-News, OWT = OpenWebText; ★ models were initialized with the pretrained RoBERTa model.

| Model | Source | # Params | Vocab. Size | Pretraining Corpora § |
|---|---|---|---|---|
| BERT | Devlin et al. [15] | 110 M | 30 K | Wikipedia, BC (16 GB) |
| RoBERTa | Liu et al. [25] | 125 M | 50 K | Wikipedia, BC, CC-News, OWT (160 GB) |
| Legal-BERT | Chalkidis et al. [16] | 110 M | 30 K | Legislation, Court Cases, Contracts (12 GB) |
| Legal-RoBERTa ★ | Geng et al. [26] | 125 M | 50 K | Patents, Court Cases (5 GB) |
| PrivBERT ★ | Srinath et al. [27] | 125 M | 50 K | Privacy policies (17 GB) |

#### 4.1.1. BERT

Proposed by Devlin et al. [15], BERT is perhaps the most well-known transformer language model. BERT utilizes the WordPiece tokenizer [28] and is case-insensitive. It is pretrained with the Masked Language Model (MLM) and Next-Sentence Prediction (NSP) tasks on the Wikipedia and BookCorpus corpora.

#### 4.1.2. RoBERTa

Liu et al. [25] proposed RoBERTa as an improvement to BERT. RoBERTa uses dynamic token masking and eliminates the NSP task during pretraining. Furthermore, it uses a case-sensitive byte-level Byte-Pair Encoding [29] tokenizer and is pretrained on larger corpora. Liu et al. [25] reported improved results on various benchmarks using RoBERTa over BERT.

#### 4.1.3. Legal-BERT

Chalkidis et al. [16] proposed Legal-BERT by pretraining BERT from scratch on legal corpora consisting of legislation, court cases and contracts. The sub-word vocabulary of Legal-BERT was learned from scratch using the SentencePiece [30] tokenizer to better support legal terminology. Legal-BERT was the best overall performing model in the LexGLUE benchmark, as reported in Chalkidis et al. [7].

#### 4.1.4. Legal-RoBERTa

Inspired by Legal-BERT, Geng et al. [26] proposed Legal-RoBERTa by further pretraining RoBERTa on legal corpora, specifically patents and court cases. Legal-RoBERTa is pretrained on less legal data than Legal-BERT while producing similar results on downstream fine-tuning legal domain tasks.

#### 4.1.5. PrivBERT

Due to the scarcity of large corpora in the privacy domain, Srinath et al. [27] proposed PrivaSeer, a novel corpus of 1M English language website privacy policies crawled from the web. They subsequently proposed PrivBERT by further pretraining RoBERTa on the PrivaSeer corpus.

### 4.2. Task-Specific Approaches

Given the aforementioned models and tasks, we now describe our task-specific fine-tuning and evaluation approaches. Given an input sequence $s = \{w_1, w_2, \ldots, w_N\}$ consisting of $N$ sequential sub-word tokens, we feed $s$ into a transformer encoder and obtain a contextual representation $\{h_0, h_1, \ldots, h_N\}$ where $h_i \in \mathbb{R}^D$ and $D$ is the output dimensionality of the transformer encoder. Here, $h_0$ refers to the contextual embedding for the starting token which is `[CLS]` for BERT-derived models and `<s>` for RoBERTa-derived models. For PolicyQA and PrivacyQA, the input sequence $s$ is composed by concatenating the question and context/answer pairs, respectively. The concatenated sequences are separated by a separator token, which is `[SEP]` for BERT-derived models and `</s>` for RoBERTa-derived models.

#### 4.2.1. Sigmoid and Softmax Functions

We utilize the sigmoid (1) and softmax (2) functions in our task-specific approaches. The sigmoid function is useful for binary-classification cases since it monotonically transforms a single logit in $\mathbb{R}$ into probability space, that is, $[0, 1]$. The softmax function performs a similar role by transforming a set of logits, each from $\mathbb{R}$ into probability space, such that the individual probabilities sum up to unity. Both functions are differentiable and are therefore useful for gradient descent techniques used in deep learning.

$$\text{sigmoid}(x) = \frac{1}{1 + e^{-x}} \in (0, 1), \quad x \in \mathbb{R} \tag{1}$$

$$\text{softmax}(\mathbf{x})_i = \frac{e^{x_i}}{\sum_{j=1}^{N} e^{x_j}} \in (0, 1), \quad \mathbf{x} = \{x_1, \ldots, x_N\} \in \mathbb{R}^N \tag{2}$$

#### 4.2.2. Sequence Classification

The $h_0$ embedding is fed into a class-wise sigmoid classifier (3) and softmax classifier (4) for multi-label and binary/multi-class tasks, respectively. The classifier has weights $W \in \mathbb{R}^{D \times C}$ and bias $b \in \mathbb{R}^C$ and is used to predict the probability vector $y \in \mathbb{R}^C$, where $C$ refers to the number of output classes. We fine-tune models end-to-end by minimizing the binary cross-entropy loss and cross-entropy loss for multi-label and binary/multi-class tasks, respectively.

$$y_i = \text{sigmoid}([W^\top h_0 + b]_i) \tag{3}$$

$$y = \text{softmax}(W^\top h_0 + b) \tag{4}$$

We report the macro- and micro-average $F_1$ scores for all sequence classification tasks, since the former ignores class imbalance while the latter takes it into account.

#### 4.2.3. Multi-Task Token Classification

Each $h_i \in \{h_1, h_2, \ldots, h_N\}$ token embedding is fed into $J$-independent softmax classifiers with weights $W_j \in \mathbb{R}^{D \times C_j}$ and bias $b_j \in \mathbb{R}^{C_j}$ to predict the token probability vector $y_{ij} \in \mathbb{R}^{C_j}$, where $C_j$ refers to the number of output BIO classes per subtask $j \in \{1, 2, \ldots, J\}$. We fine-tune models end-to-end by minimizing the cross-entropy loss across all tokens and subtasks.

$$y_{ij} = \text{softmax}(W_j^\top h_i + b_j) \tag{5}$$

We report the macro- and micro-average $F_1$ scores for all multi-task token classification tasks by averaging the respective metrics for each subtask. Furthermore, we ignore cases where B or I prefixes are mismatched as long as the main token class is correct.

#### 4.2.4. Reading Comprehension

Each $h_i \in \{h_1, h_2, \ldots, h_N\}$ token embedding is fed into two independent linear layers with weights $W_j \in \mathbb{R}^D$ and bias $b_j \in \mathbb{R}$ where $j \in \{1, 2\}$. These linear outputs are then

concatenated per layer and a softmax function is applied to form a probability vector $y_j$ across all tokens for answer-start and answer-end token probabilities, respectively. We fine-tune models end-to-end by minimizing the cross-entropy loss on the gold answer-start and answer-end indices.

$$y_j = \text{softmax}\Big( \begin{bmatrix} W_j \cdot h_1 + b_j & \dots & W_j \cdot h_N + b_j \end{bmatrix} \Big) \qquad (6)$$

Similar to SQuAD [24], we report the sample $F_1$ and exact match accuracy for our reading comprehension task. It is worth noting that Rajpurkar et al. [24] refer to their reported $F_1$ score as a macro-average, whereas we refer to it as the sample-average as we believe this is a more accurate term.

*4.3. Benchmark Configuration*

We run PrivacyGLUE benchmark tasks with the following configuration:

- We train all models for 20 epochs with a batch size of 16. We utilize a linear learning rate scheduler with a warmup ratio of 0.1 and peak learning rate of $3 \times 10^{-5}$. We utilize AdamW [31] as our optimizer and use mixed 16-bit float precision for more efficient training. Finally, we monitor respective metrics on the validation datasets and utilize early stopping if the validation metric does not improve for five epochs.
- We use `Python v3.8.13`, `CUDA v11.7`, `PyTorch v1.12.1` [32] and `Transformers v4.19. 4` [33] as our core software dependencies.
- We use the following HuggingFace model tags: `bert-base-uncased`, `roberta-base`, `nlpaueb/legal-bert-base-uncased`, `saibo/legal-roberta-base`, `mukund/privbert` for BERT, RoBERTa, Legal-BERT, Legal-RoBERTa and PrivBERT, respectively.
- We use 10 random seeds for each benchmark run, that is, $\{0, 1, 2, 3, 4, 5, 6, 7, 8, 9\}$. This provides a distribution of results that can be used for statistical significance testing.
- We run the PrivacyGLUE benchmark on a Lambda workstation with $4 \times$ NVIDIA RTX A4000 (16 GB VRAM) GPUs, 125 GB RAM and Intel i9-10920X CPU (12 cores) for $\sim$180 h.
- We use `Weights and Biases v 0.13.3` [34] to monitor model metrics during training and for intermediate report generation.

## 5. Results

After running the PrivacyGLUE benchmark with 10 random seeds, we collect results on the test-sets of all tasks. Figure 3 shows the respective results in a graphical form, while Table A3 in Appendix C shows the numerical results in a tabular form. In terms of absolute metrics, we observe that PrivBERT outperforms other models for all PrivacyGLUE tasks. We apply the Mann–Whitney U-test [35] over random seed metric distributions and find that PrivBERT significantly outperforms other models on six out of seven PrivacyGLUE tasks with $p \leq 0.05$, where Policy-Detection was the task where the significance threshold was not met. We utilize the Mann–Whitney U-test because it does not require a normal distribution for test-set metrics, an assumption which has not been extensively validated for deep neural networks [36].

In Figure 3, we observe large differences between the two representative metrics for OPP-115, Policy-Detection, PolicyIE-A, PrivacyQA and PolicyQA. For the first four of the aforementioned tasks, this is because of data imbalance resulting in the micro-average $F_1$ being significantly higher since it can be skewed by the metric of the majority class. For PolicyQA, this occurs because the EM metric requires exact matches and is therefore much stricter than the sample $F_1$ metric. Furthermore, we observe an exceptionally large standard deviation on PI-Extract metrics compared to other tasks. This can be attributed to data imbalance between the four subtasks of PI-Extract, with the `NOT_COLLECT` and `NOT_SHARE` subtasks having less than 100 total examples each.

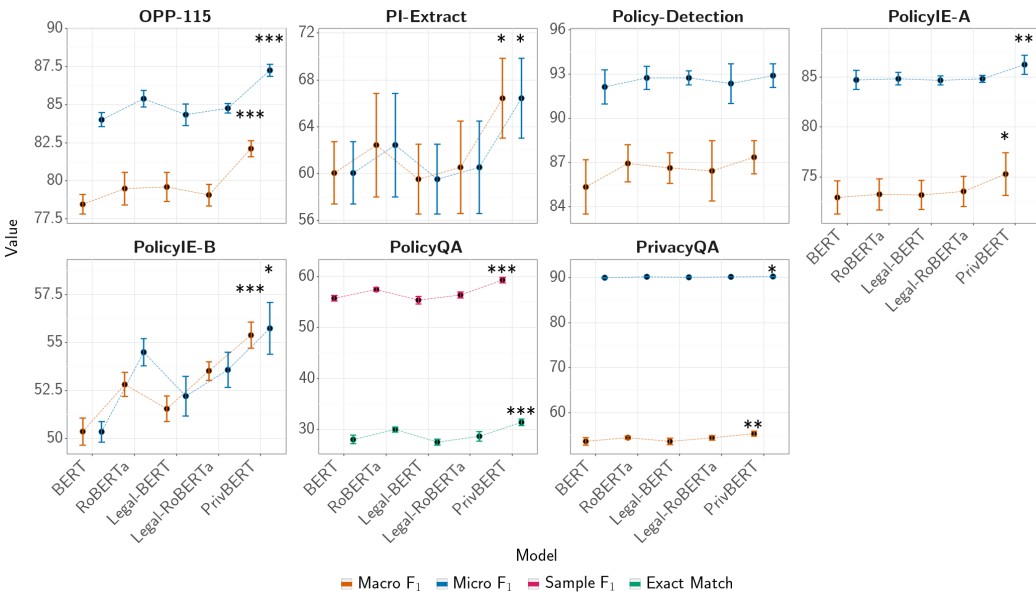

**Figure 3.** Test-set results of the PrivacyGLUE benchmark where points indicate mean performance and error bars indicate standard deviation over 10 random seeds; *** implies $p \leq 0.001$ , ** implies $0.001 < p \leq 0.01$, * implies $0.01 < p \leq 0.05$ given an alternative hypothesis that PrivBERT has a greater performance metric than all other models in a task using the Mann–Whitney U-test.

We apply the arithmetic, geometric and harmonic means to aggregated metric means and standard deviations, as shown in Table 3. With this, we observe the following general ranking of models from best to worst: PrivBERT, RoBERTa, Legal-RoBERTa, Legal-BERT and BERT. Interestingly, models derived from RoBERTa generally outperformed models derived from BERT. Using the arithmetic mean for simplicity, we observe that PrivBERT outperforms all other models by 2–3%. As an additional point, we utilize the aggregated metrics in Table 3 to rank the central tendencies of model performances.

**Table 3.** Macro-aggregation of means ($\mu$) and standard deviations ($\sigma$) per model using the arithmetic mean (A-Mean), geometric mean (G-Mean) and harmonic mean (H-Mean).

| Model | A-Mean | | G-Mean | | H-Mean | |
|---|---|---|---|---|---|---|
| | $\mu$ | $\sigma$ | $\mu$ | $\sigma$ | $\mu$ | $\sigma$ |
| BERT | 67.5 | 1.1 | 64.6 | 0.9 | 61.1 | 0.6 |
| RoBERTa | 69.0 | 1.2 | 66.4 | 0.7 | 63.2 | 0.3 |
| Legal-BERT | 67.9 | 1.1 | 64.9 | 0.8 | 61.2 | 0.4 |
| Legal-RoBERTa | 68.5 | 1.3 | 65.7 | 0.8 | 62.3 | 0.4 |
| PrivBERT | **70.8** | 1.2 | **68.3** | 0.8 | **65.2** | 0.5 |

## 6. Discussion

With the PrivacyGLUE benchmark results, we revisit our privacy vs. legal language domain claim from Section 1 and discuss our model–pair agreement analysis for detecting PrivacyGLUE task examples where models benefited from domain specialization.

### 6.1. Privacy vs. Legal Language Domain

We initially provided evidence from Figure 1 suggesting that the privacy language domain is distinct from the legal language domain. We believe that our PrivacyGLUE results further support this initial claim. If the privacy language domain was subsumed under the legal language domain, we could have observed Legal-RoBERTa and Legal-BERT performing competitively with PrivBERT. Instead, we observed that the legal models

underperformed compared to both PrivBERT and RoBERTa, further indicating that the privacy language domain is distinct and requires its own NLP benchmark.

*6.2. Model–Pair Agreement Analysis*

PrivBERT, the top-performing model, differentiates itself from other models by its in-domain pretraining on the PrivaSeer corpus [27]. Therefore, we can infer that PrivBERT incorporated *knowledge* of privacy policies through its pretraining and became specialized for fine-tuning tasks in the privacy language domain. We investigate this specialization using model–pair agreement analysis to detect examples where PrivBERT had a competitive advantage over other models. Consequently, we detect examples where PrivBERT was disadvantaged due to its in-domain pretraining.

We compare $10 \times 10 = 100$ random seed combinations for all test-set pairs between PrivBERT and other models. Each prediction-pair can be classified into one of four mutually exclusive categories (B, P, O and N) shown below. Categories B and N represent examples that are either not challenging or too challenging for both PrivBERT and the other model respectively. Categories P and O are more interesting for us since they indicate examples where PrivBERT had a competitive advantage and disadvantage over the other model, respectively. Therefore, we focus on categories P and O in our analysis. We classify examples over all random seed combinations and take the majority occurrence for each category within its distribution.

**Category B:** Both PrivBERT and the other model were correct, that is, (PrivBERT, Other Model)

**Category P:** PrivBERT was correct and the other model was wrong, that is, (PrivBERT, ¬ Other Model)

**Category O:** Other model was correct and PrivBERT was wrong, i.e., (¬ PrivBERT, Other Model)

**Category N:** Neither PrivBERT nor the other model was correct, that is, (¬ PrivBERT, ¬ Other Model)

Figure 4 shows a relative distribution of majority categories across model–pairs and PrivacyGLUE tasks. We observe that category P is always greater than category O, which correlates with PrivBERT outperforming all other models. We also observe that category P is often the greatest when compared against BERT, implying that PrivBERT has the most competitive advantage over BERT. Surprisingly, we also observe category O is often the greatest when compared against BERT, implying that BERT has the highest absolute advantage over PrivBERT. This is an insightful observation since we would have expected BERT to have the least competitive advantage given its lowest overall PrivacyGLUE performance.

To investigate PrivBERT's competitive advantage and disadvantage against BERT, we extract several examples from categories P and O in the PrivacyQA task for brevity. Two interesting examples are listed in Table 4 and additional examples can be found in Table A4 in Appendix D. From Table 4, we speculate that PrivBERT specializes in example 1978 because it contains several privacy-specific terms such as "third parties" and "explicit consent". On the other hand, we speculate that BERT specializes in example 33237 since it contains more generic information regarding encryption and SSL, which also happens to be a topic in BERT's Wikipedia pretraining corpus as seen in Figure 1 and Table 2.

Looking at further examples in Table A4, we can also observe that all sampled category P examples have the `Relevant` label, while many sampled category O examples have the `Irrelevant` label. On further analysis of the PrivacyQA test-set, we find that 71% of category P examples have the `Relevant` label and 61% of category O samples have the `Irrelevant` label. We can infer that PrivBERT specializes in the minority `Relevant` label while BERT specializes in the majority `Irrelevant` label as the former label could require more privacy knowledge than the latter.

**Table 4.** Test-set examples from PrivacyQA that fall under categories P and O for PrivBERT vs. BERT.

| Category P | Category O |
|---|---|
| **ID:** 1978<br>**Question:** Who can see my information?<br>**Answer:** We do not sell or rent your personal information to third parties for their marketing purposes without your explicit consent.<br>**Label:** Relevant | **ID:** 33237<br>**Question:** Could the wordscapes app contain malware?<br>**Answer:** We encrypt the transmission of all information using secure socket layer technology (SSL).<br>**Label:** Relevant |

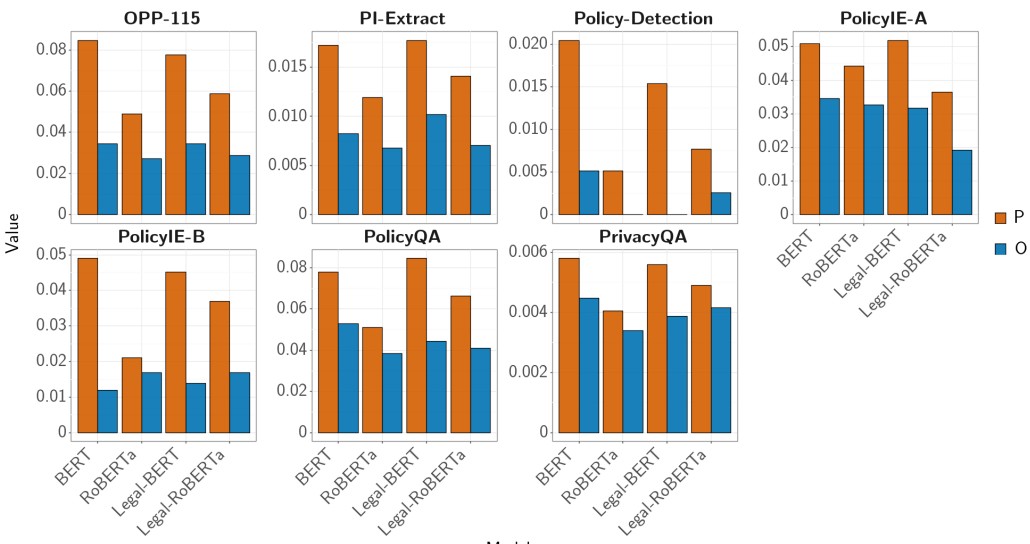

**Figure 4.** Model–pair agreement analysis of PrivBERT against other models over all PrivacyGLUE tasks; bars represent proportions of examples per model–pair and task which fell into categories P and O; all models on the x-axis are compared against PrivBERT.

## 7. Conclusions and Further Work

In this paper, we described the importance of data privacy in modern digital life and observed the lack of an NLP benchmark in the privacy language domain despite its distinctness. To address this, we proposed PrivacyGLUE as the first comprehensive benchmark for measuring general language understanding in the privacy language domain. We released benchmark performances from the BERT, RoBERTa, Legal-BERT, Legal-RoBERTa and PrivBERT transformer language models. Our findings showed that PrivBERT outperforms other models by an average of 2–3% over all PrivacyGLUE tasks, shedding light on the importance of in-domain pretraining for privacy policies. We applied model–pair agreement analysis to detect PrivacyGLUE examples where PrivBERT's pretraining provides a competitive advantage and disadvantage. By benchmarking holistic model performances, we believe PrivacyGLUE can accelerate NLP research into the privacy language domain and ultimately improve general language understanding of privacy policies for both humans and AI algorithms. Ultimately, this will support practitioners in selecting the best models to use in applications that simplify privacy policies. An example of such an application could be a browser plugin that actively condenses privacy policies into their most important parts before presenting them to the user for their consent.

Looking forward, we envision several ways to further enhance our study. Firstly, we intend to apply deep-learning explainability techniques, such as Integrated Gradients [37] on examples from Table 4, to explore PrivBERT's and BERT's token-level attention attributions for categories P and O. Additionally, we intend to benchmark large prompt-based transformer language models such as T5 [38] and T0 [39], as they incorporate large amounts of knowledge from the various sequence-to-sequence tasks that they were trained on. Fi-

nally, we plan to continue maintaining our PrivacyGLUE GitHub repository and host new model results from the community.

## 8. Limitations

To the best of our knowledge, our study has two main limitations. While we provide performances from transformer language models, our study does not provide human expert performances on PrivacyGLUE. This would have been a valuable contribution to judge how competitive language models are against human expertise. However, this limitation can be challenging to address due to the difficulty in finding experts and high costs for their services. Additionally, our study only focuses on English language privacy tasks and omits multilingual scenarios.

## 9. Ethics Statement

In this section, we provide an overview of ethical considerations taken into account in our study. These include original work attribution, social impact and software licensing.

### 9.1. Original Work Attribution

All datasets used to compose PrivacyGLUE are publicly available and originate from previous studies. We cite these studies in our paper and include references for them in our GitHub repository. Furthermore, we clearly illustrate how these datasets were used to form the PrivacyGLUE benchmark.

### 9.2. Social Impact

PrivacyGLUE could be used to produce fine-tuned transformer language models, which could then be utilized in downstream applications to help users understand privacy policies and/or answer questions regarding them. We believe this could have a positive social impact as it would empower users to better understand lengthy and complex privacy policies. That being said, application developers should perform appropriate risk analyses when using fine-tuned transformer language models. Important points to consider include the varying performance ranges on PrivacyGLUE tasks and known examples of implicit bias, such as gender and racial bias, that transformer language models incorporate through their large-scale pretraining [40].

### 9.3. Software Licensing

We release the source code for PrivacyGLUE under version 3 of the GNU General Public License (GPL-3.0) with all datasets retaining their original licenses, which could differ from GPL-3.0. We chose GPL-3.0 as it is a strong copyleft license that protects user freedoms such as the freedom to use, modify and distribute software.

**Author Contributions:** Conceptualization, A.S., A.W., C.B., M.A.R. and L.M.; methodology, A.S., C.B., M.A.R., A.W. and L.M.; software, A.S. and C.B.; validation, A.S., C.B., M.A.R., A.W. and L.M.; formal analysis, A.S. and C.B.; investigation, A.S., C.B., M.A.R., A.W. and L.M.; resources, A.S. and A.W.; data curation, A.S., C.B. and A.W.; writing—original draft preparation, A.S., C.B., M.A.R., A.W. and L.M.; writing—review and editing, A.S., C.B., M.A.R., A.W. and L.M.; visualization, A.W., A.S. and C.B.; supervision, L.M. and A.S.; project administration, L.M.; funding acquisition, L.M. All authors have read and agreed to the published version of the manuscript.

**Funding:** This research received no external funding.

**Institutional Review Board Statement:** Not applicable.

**Informed Consent Statement:** Not applicable.

**Data Availability Statement:** The source code and the data used in the PrivacyGLUE benchmark (while not already publicly provided by other sources) is freely avaiable at the GitHub repository: https://github.com/infsys-lab/privacy-glue (accessed on 22 January 2023).

**Conflicts of Interest:** The authors declare no conflict of interest.

## Abbreviations

The following abbreviations are used in this manuscript:

| | |
|---|---|
| UN | United Nations |
| AI | Artificial Intelligence |
| NLP | Natural Language Processing |
| GLUE | General Language Understanding Evaluation benchmark |
| SuperGLUE | Super General Language Understanding Evaluation benchmark |
| SOTA | State of the Art |
| LexGLUE | Legal General Language Understanding Evaluation benchmark |
| EURLEX | European Legal Texts (Portal) |
| BERT | Bidirectional Encoder Representations from Transformers |
| UMAP | Uniform Manifold Approximation and Projection |
| PrivacyGLUE | Privacy General Language Understanding Evaluation benchmark |
| OPP-115 | Online Privacy Policies, set of 115 |
| PI-Extract | Personal Information Extraction |
| PolicyIE | Policy Intent Extraction |
| PolicyQA | Policy Questions and Answers |
| PrivacyQA | Privacy Questions and Answers |
| RoBERTa | Robustly Optimized BERT Pretraining Approach |
| GEM | Generation, Evaluation and Metrics |
| XTREME | Cross-Lingual Transfer Evaluation of Multilingual Encoders |
| XTREME-R | Cross-Lingual Transfer Evaluation of Multilingual Encoder Revisited |
| BIO | Beginning, Inside and Outside |
| NER | Named Entity Recognition |
| SQuAD | Stanford Question Answering Dataset |
| T5 | Text-To-Text Transfer Transformer |
| T0 | T5 for zero-shot task generalization |

## Appendix A. Detailed Label Information

**Table A1.** Breakdown of labels for each PrivacyGLUE task; PolicyQA is omitted from this table since it is a reading comprehension task and does not have explicit labels like other tasks.

| Task | Labels |
|---|---|
| OPP-115 | Data Retention, Data Security, Do Not Track, First Party Collection/Use, International and Specific Audiences Introductory/Generic, Policy Change, Practice not covered, Privacy contact information, Third Party Sharing/Collection, User Access, Edit and Deletion, User Choice/Control |
| PI-Extract | **Subtask-I:** {B,I}-COLLECT, O<br>**Subtask-II:** {B,I}-NOT_COLLECT, O<br>**Subtask-III:** {B,I}-NOT_SHARE, O<br>**Subtask-IV:** {B,I}-SHARE, O |
| Policy-Detection | Not Policy, Policy |
| PolicyIE-A | Other, data-collection-usage, data-security-protection, data-sharing-disclosure, data-storage-retention-deletion |
| PolicyIE-B | **Subtask-I:** {B,I}-data-protector, {B,I}-data-protected, {B,I}-data-collector, {B,I}-data-collected, {B,I}-data-receiver, {B,I}-data-retained, {B,I}-data-holder, {B,I}-data-provider, {B,I}-data-sharer, {B,I}-data-shared, storage-place, {B,I}-retention-period, {B,I}-protect-against, {B,I}-action, O<br>**Subtask-II:** {B,I}-purpose-argument, {B,I}-polarity, {B,I}-method, {B,I}-condition-argument, O |
| PrivacyQA | Irrelevant, Relevant |

## Appendix B. PrivacyGLUE Task Examples

**Table A2.** Representative examples of each PrivacyGLUE benchmark task.

| Task | Input | Target |
|------|-------|--------|
| OPP-115 | Revision Date: 24 March 2015 | `Introductory/Generic,`<br>`Policy Change` |
| PI-Extract | We may collect and share your IP address but not your email address with our business partners. | **Subtask-I:** `O O O O O B-COLLECT I-COLLECT I-COLLECT O O O O O O O O O`<br>**Subtask-II:** `O O O O O O O O O B-NOT_COLLECT I-NOT_COLLECT I-NOT_COLLECT O O O O O`<br>**Subtask-II:** `O O O O O O O O O O B-NOT_SHARE I-NOT_SHARE I-NOT_SHARE O O O O O`<br>**Subtask-IV:** `O O O O O B-SHARE I-SHARE I-SHARE O O O O O O O O O` |
| Policy-Detection | Log in through another service: * Facebook * Google | `Not Policy` |
| PolicyIE-A | To backup and restore your Pocket AC camera log | `data-collection-usage` |
| PolicyIE-B | Access to your personal information is restricted. | **Subtask-I:** `O O B-data-provider B-data-protected I-data-protected O B-action O`<br>**Subtask-II:** `B-method O O O O O O` |
| PolicyQA | **Question:** How do they secure my data?<br>**Context:** Users can visit our site anonymously | **Answer:** Users can visit our site anonymously |
| PrivacyQA | **Question:** What information will you collect about my usage?<br>**Answer:** Location information | `Relevant` |

## Appendix C. PrivacyGLUE Benchmark Results

**Table A3.** Test-set results of the PrivacyGLUE benchmark; [♯] m-$F_1$ refers to macro-average $F_1$, μ-$F_1$ refers to the micro-average $F_1$, s refers to sample-average $F_1$, EM refers to the exact match accuracy, metrics are reported as percentages with the following format: mean$_{\pm \text{standard deviation}}$.

| Task | Metric [♯] | BERT | RoBERTa | Legal-BERT | Legal-RoBERTa | PrivBERT |
|------|-----------|------|---------|------------|---------------|----------|
| OPP-115 | m-$F_1$ | $78.4_{\pm 0.6}$ | $79.5_{\pm 1.1}$ | $79.6_{\pm 1.0}$ | $79.1_{\pm 0.7}$ | $\mathbf{82.1}_{\pm 0.5}$ |
|  | μ-$F_1$ | $84.0_{\pm 0.5}$ | $85.4_{\pm 0.5}$ | $84.3_{\pm 0.7}$ | $84.7_{\pm 0.3}$ | $\mathbf{87.2}_{\pm 0.4}$ |
| PI-Extract | m-$F_1$ | $60.0_{\pm 2.7}$ | $62.4_{\pm 4.4}$ | $59.5_{\pm 3.0}$ | $60.5_{\pm 3.9}$ | $\mathbf{66.4}_{\pm 3.4}$ |
|  | μ-$F_1$ | $60.0_{\pm 2.7}$ | $62.4_{\pm 4.4}$ | $59.5_{\pm 3.0}$ | $60.5_{\pm 3.9}$ | $\mathbf{66.4}_{\pm 3.4}$ |
| Policy-Detection | m-$F_1$ | $85.3_{\pm 1.8}$ | $86.9_{\pm 1.3}$ | $86.6_{\pm 1.0}$ | $86.4_{\pm 2.0}$ | $\mathbf{87.3}_{\pm 1.1}$ |
|  | μ-$F_1$ | $92.1_{\pm 1.2}$ | $92.7_{\pm 0.8}$ | $92.7_{\pm 0.5}$ | $92.4_{\pm 1.3}$ | $\mathbf{92.9}_{\pm 0.8}$ |
| PolicyIE-A | m-$F_1$ | $72.9_{\pm 1.7}$ | $73.2_{\pm 1.6}$ | $73.2_{\pm 1.5}$ | $73.5_{\pm 1.5}$ | $\mathbf{75.3}_{\pm 2.2}$ |
|  | μ-$F_1$ | $84.7_{\pm 1.0}$ | $84.8_{\pm 0.6}$ | $84.7_{\pm 0.5}$ | $84.8_{\pm 0.3}$ | $\mathbf{86.2}_{\pm 1.0}$ |
| PolicyIE-B | m-$F_1$ | $50.3_{\pm 0.7}$ | $52.8_{\pm 0.6}$ | $51.5_{\pm 0.7}$ | $53.5_{\pm 0.5}$ | $\mathbf{55.4}_{\pm 0.7}$ |
|  | μ-$F_1$ | $50.3_{\pm 0.5}$ | $54.5_{\pm 0.7}$ | $52.2_{\pm 1.0}$ | $53.6_{\pm 0.9}$ | $\mathbf{55.7}_{\pm 1.3}$ |
| PolicyQA | s-$F_1$ | $55.7_{\pm 0.5}$ | $57.4_{\pm 0.4}$ | $55.3_{\pm 0.7}$ | $56.3_{\pm 0.6}$ | $\mathbf{59.3}_{\pm 0.5}$ |
|  | EM | $28.0_{\pm 0.9}$ | $30.0_{\pm 0.5}$ | $27.5_{\pm 0.6}$ | $28.6_{\pm 0.9}$ | $\mathbf{31.4}_{\pm 0.6}$ |
| PrivacyQA | m-$F_1$ | $53.6_{\pm 0.8}$ | $54.4_{\pm 0.3}$ | $53.6_{\pm 0.8}$ | $54.4_{\pm 0.5}$ | $\mathbf{55.3}_{\pm 0.6}$ |
|  | μ-$F_1$ | $90.0_{\pm 0.1}$ | $90.2_{\pm 0.0}$ | $90.0_{\pm 0.1}$ | $90.2_{\pm 0.1}$ | $\mathbf{90.2}_{\pm 0.1}$ |

## Appendix D. Additional PrivacyQA Examples from Categories P and O

**Table A4.** Additional test-set examples from PrivacyQA that fall under categories P and O for PrivBERT vs. BERT; note that these examples are not paired and can therefore be compared in any order between categories.

| Category P | Category O |
|---|---|
| **ID:** 9227<br>**Question:** Will the app use my data for marketing purposes?<br>**Answer:** We will never share with or sell the information gained through the use of Apple HealthKit, such as age, weight and heart rate data, to advertisers or other agencies without your authorization.<br>**Label:** Relevant | **ID:** 8749<br>**Question:** Will my fitness coach share my information with others?<br>**Answer:** Develop new services.<br>**Label:** Irrelevant |
| **ID:** 10858<br>**Question:** What information will this app have access to of mine?<br>**Answer:** Information you make available to us when you open a Keep account, as set out above;<br>**Label:** Relevant | **ID:** 47271<br>**Question:** Who will have access to my medical information?<br>**Answer:** 23andMe may share summary statistics, which do not identify any particular individual or contain individual-level information, with our qualified research collaborators.<br>**Label:** Irrelevant |
| **ID:** 18704<br>**Question:** Does it share my personal information with others?<br>**Answer:** We may also disclose Non-Identifiable Information:<br>**Label:** Relevant | **ID:** 54904<br>**Question:** What data do you keep and for how long?<br>**Answer:** We may keep activity data on a non-identifiable basis to improve our services.<br>**Label:** Irrelevant |
| **ID:** 45935<br>**Question:** Will my test results be shared with any third party entities?<br>**Answer:** 23andMe may share summary statistics, which do not identify any particular individual or contain individual-level information, with our qualified research collaborators.<br>**Label:** Relevant | **ID:** 57239<br>**Question:** Do you sell any of our data?<br>**Answer:** (c) Advertising partners: to enable the limited advertisements on our service, we may share a unique advertising identifier that is not attributable to you, with our third party advertising partners, and advertising service providers, along with certain technical data about you (your language preference, country, city, and device data), based on our legitimate interest.<br>**Label:** Relevant |
| **ID:** 50467<br>**Question:** Can I delete my personally identifying information?<br>**Answer:** (Account Deletion), we allow our customers to delete their accounts at any time.<br>**Label:** Relevant | **ID:** 59334<br>**Question:** Does the app protect my account details from being accessed by other people?<br>**Answer:** Note that chats with bots and Public Accounts, and communities are not end-to-end encrypted, but we do encrypt such messages when sent to the Viber servers and when sent from the Viber servers to the third party (the Public Account owner and/or additional third party tool (e.g., CRM solution) integrated by such owner).<br>**Label:** Irrelevant |

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
