# Peer review of "PrivacyGLUE: A Benchmark Dataset for General Language Understanding in Privacy Policies"

_applsci, doi:10.3390/app13063701_

Round 1

Reviewer 1 Report

The authors have proposed PrivacyGLUE as the first comprehensive benchmark for measuring general language understanding in the privacy language domain. However, some parts are not well understood from the document. My comments are as follows:

1.     In general, the Introduction should more strongly motivate why the authors feel their proposed is needed. Please add and supplement the necessity and feasibility of the proposed PrivacyGLUE benchmark.

2.     The six datasets and five models are described in detail at length in 3 and 4.1 respectively. But I'd rather know what the criteria and basis are for selecting the datasets or models.

3.     What are the software and hardware environments for the experimental setup in the manuscript? Please add the relevant explanation.

4.     The authors have proposed PrivacyGLUE as the first comprehensive benchmark for measuring general language understanding in the privacy language domain. However, the models and task-specific approaches in this paper are referred to the existing references, so what is the innovation of this paper? It would be more satisfying if some innovations exist in some step/steps.

5.     The arithmetic, geometric and harmonic means are applied to aggregated metric means and standard deviations as shown in Table 3. The authors mentioned that the following general ranking of models from best to worst are PrivBERT, RoBERTa, Legal-RoBERTa, Legal-BERT and BERT. I don't think that's a very precise summary, because from the standard deviation analysis, PrivBERT has a higher standard deviation than other methods, like BERT or Legal-BERT. Please explain the reason for the higher standard deviation.

6.     It will be more satisfying if the authors can supplement the flow chart of experimental design and implementation.

Reviewer 3 Report

Please see atatched detailed review 

Reviewer 4 Report

The authors analyse some language domain and use different benchmark and different Natural Language Processing. There are a lot of acronyms and terms in this paper (GEM benchmark, SOTA, etc.). I recomand to say few words about each one (details at first time of use it)  for an easier follow-up of the context. Even if they are established names, the abundance of their use determines a tedious aspect of the article. Relation (1) to (4) use softmax and sigmoid classifier. Softmax converts them to values between 0 and 1 so that they can be interpreted as probabilities, but this do not appear as explained throughout the article (for readingability). This is not necessary for widely used formulas, like standard deviations, arithmetic mean, geometric mean and harmonic mean, etc. Results are clear presented.

Round 2

Reviewer 1 Report

The authors have carefully improved the manuscript according to the reviewer's comments.